

# An automated method to enrich consumer health vocabularies using GloVe word embeddings and an auxiliary lexical resource

Mohammed Ibrahim, Susan Gauch, Omar Salman and Mohammed Alqahtani

Computer Science and Computer Engineering, University of Arkansas at Fayetteville, Fayetteville, AR, United States

## ABSTRACT

**Background:** Clear language makes communication easier between any two parties. A layman may have difficulty communicating with a professional due to not understanding the specialized terms common to the domain. In healthcare, it is rare to find a layman knowledgeable in medical terminology which can lead to poor understanding of their condition and/or treatment. To bridge this gap, several professional vocabularies and ontologies have been created to map laymen medical terms to professional medical terms and vice versa.

**Objective:** Many of the presented vocabularies are built manually or semi-automatically requiring large investments of time and human effort and consequently the slow growth of these vocabularies. In this paper, we present an automatic method to enrich laymen's vocabularies that has the benefit of being able to be applied to vocabularies in any domain.

**Methods:** Our entirely automatic approach uses machine learning, specifically Global Vectors for Word Embeddings (GloVe), on a corpus collected from a social media healthcare platform to extend and enhance consumer health vocabularies. Our approach further improves the consumer health vocabularies by incorporating synonyms and hyponyms from the WordNet ontology. The basic GloVe and our novel algorithms incorporating WordNet were evaluated using two laymen datasets from the National Library of Medicine (NLM), Open-Access Consumer Health Vocabulary (OAC CHV) and MedlinePlus Healthcare Vocabulary.

**Results:** The results show that GloVe was able to find new laymen terms with an F-score of 48.44%. Furthermore, our enhanced GloVe approach outperformed basic GloVe with an average F-score of 61%, a relative improvement of 25%. Furthermore, the enhanced GloVe showed a statistical significance over the two ground truth datasets with $P < 0.001$.

**Conclusions:** This paper presents an automatic approach to enrich consumer health vocabularies using the GloVe word embeddings and an auxiliary lexical source, WordNet. Our approach was evaluated used healthcare text downloaded from *MedHelp.org*, a healthcare social media platform using two standard laymen vocabularies, OAC CHV, and MedlinePlus. We used the WordNet ontology to expand the healthcare corpus by including synonyms, hyponyms, and hypernyms for each layman term occurrence in the corpus. Given a seed term selected from a concept in the ontology, we measured our algorithms' ability to automatically extract synonyms for those terms that appeared in the ground truth concept. We found

Corresponding author
Mohammed Ibrahim, mohammad.alaubeedy@gmail.com

that enhanced GloVe outperformed GloVe with a relative improvement of 25% in the F-score.

## INTRODUCTION

An ontology is a formal description and representation of concepts with their definitions, relations, and classifications in a specific or general domain of discourse (*Grüninger & Fox, 1995*). It can decrease terminological and conceptual confusion between system software components and facilitate interoperability. Examples of ontologies in different domains are the BabelNet (*Navigli & Ponzetto, 2012*), Arabic Ontology (*Jarrar, 2011*), WordNet (*Miller, 1995*), and Gene Ontology (*Consortium, 2007*). Ontologies have been used in many domains such as document indexing (*Lipscomb, 2000*), personalizing user's profiles for information retrieval systems (*Gauch, Chaffee & Pretschner, 2003*; *Pretschner & Gauch, 1999*; *Trajkova & Gauch, 2004*; *Chaffee & Gauch, 2000*; *Challam, Gauch & Chandramouli, 2007*; *Challam & Gauch, 2004*), and providing readable data for semantic web applications (*Maedche & Staab, 2001*; *Fensel et al., 2001*; *Doan et al., 2002*; *McIlraith, Son & Zeng, 2001*).

Several ontologies have been developed and/or proposed for the healthcare domain. One of the biggest healthcare ontologies in the field of biomedicine is the Unified Medical Language system (UMLS). This ontology consists of more than 3,800,000 professional biomedicine concepts. It lists biomedical concepts from different resources, including their part of speech and variant forms (*Bodenreider, 2004*). The National Library of Medicine (NLM) manages the UMLS ontology and updates it yearly. As examples of the professional vocabularies included in the UMLS, the Gene Ontology (GO) (*Consortium, 2007*), Disease Ontology (DO) (*Schriml et al., 2011*), and Medical Subject Headings (MeSH) (*Lipscomb, 2000*). The UMLS not only has professional vocabularies but also included laymen vocabularies. These vocabularies provide straightforward terms mapped to the professional medical concepts.

With the advancement of medical technology and the emergence of internet social media, people are more connected than before. In terms of medical technology, there are many efforts to build smart devices that can interact and provide health information. On social media, people started not only sharing their climate concerns, politics, or social problems, but also their health problems. The Pew Research Center conducted a telephone survey in 2010 and reported that 80% of the United States internet users looked for a healthcare information. The survey showed that 66% of those users looked for a specific disease or medical issue and roughly 55% of them looked for a remedies treat to their medical problems (*Fox, 2011*). Another study showed that the rate of using social media by physicians grew from 41% in 2010 to 90% in 2011 (*Bosslet et al., 2011*; *George, Rovniak & Kraschnewski, 2013*; *Modahl, Tompsett & Moorhead, 2011*). In all these cases, any

retrieval system will not be able to interact effectively with laypeople unless they have a lexical source or ontology that defines the medical terminology.

Medical professionals are well-versed in specialized medical terminology developed to be a precise way for healthcare professionals to communicate with each other. However, this medical jargon is obscure to laymen and may require patients to ask for more details to be sure that they understand their condition and treatment plans (*Josh, 2017*). A recent study (*Papadakos et al., 2021*) showed the effect of health literacy on the accuracy of the information laymen are seeking related to coronavirus (COVID-19) and cancer. The study concluded that much of the cancer and COVID-19 information available does match with patients' health literacy because much of the present information has been written using professional terminology which is hard for laymen to understand. Having a way to map professional medical terminology to easier to understand laymen terms could close the communication gap between patients and the healthcare professional.

Recently, steps have been taken to close the gap between the vocabulary the professionals use in healthcare and what laymen use. It was reported by *Blanchard (2018)* that approximately five million doctor letters are sent to patients each month. Using words like *liver* instead of *hepatic* and *brain* instead of *cerebral* could make the doctor's letters much easier to laymen (*BBC News, 2018*). Thus, the Academy of Medical Royal Colleges started an initiative in 2017 in which the doctors asked to write to patients directly using plain English instead of medical terminology (*Yeginsu, 2018*). The twentieth century witnessed steps of building a lot of vocabularies that maps professional medical concepts to their laymen terms and vice versa. These vocabularies are commonly known as consumer health vocabularies. Many of these presented vocabularies are built manually or semi-automatically requiring large investments of time and human effort and consequently these vocabularies grow very slowly. In this paper, we present an automatic method to enrich laymen's vocabularies that has the benefit of being able to be applied to vocabularies in any domain.

## Consumer health vocabularies

Consumer health vocabularies can decrease the gap between laymen and professional language and help humans and machines to understand both languages. *Zeng et al. (2001)* reported poor search results when a layman searched for the term *heart attack* because physicians discuss that condition using the professional concept *myocardial infarction*. There are many laymen vocabularies proposed in the field of biomedicine such as the Open-Access and Collaborative Consumer Health Vocabulary (OAC CHV) (*Doing-Harris & Zeng-Treitler, 2011*), and MedlinePlus topics (*Miller, Lacroix & Backus, 2000*). Those laymen vocabularies should grow from time to time to cover new terms proposed by the laypeople to keep system using them updated.

Our research studies two common English laymen vocabularies, the Open-Access and Collaborative Consumer Health Vocabulary (OAC CHV) and the MedlinePlus vocabularies. The National Library of Medicine (NLM) integrated these vocabularies to the UMLS ontology. Roughly 56,000 UMLS concepts were mapped to OAC CHV concepts. Many of these medical concepts have more than one associated layman term.

**Table 1 Sample UMLS concepts and some of their OA CHV associated laymen terms.**

| CUI | UMLS Concept | Associated laymen terms | | | |
|-----|--------------|-------------------------|--|--|--|
| C0018681 | Headache | headache | headaches | head ache | ache head |
| C0003864 | Arthritis | arthritis | arthritides | arthritide | |
| C0033860 | Psoriasis | Psoriasis | psoriasi | psoriasis | |

This vocabulary has been updated many times to include new terms, and the last update was in 2011 (*Doing-Harris & Zeng-Treitler, 2011*). All of the updates incorporated human evaluation of the laymen terms before adding them to the concept. We did an experiment using all 56,000 UMLS concepts that had associated OAC CHV laymen terms. We stemmed, downcased, and removed all stopwords, punctuations, and numbers from the concepts and their laymen terms. We then compared the tokens in the concept names with the set of laymen terms. From this experiment, we found that 48% of the included laymen terms were just morphological variations of the professional medical terms with minor changes such as replacing lowercase/uppercase letters, switching between plural/single word forms, or adding numbers or punctuation. Table 1 shows a few examples of the laymen and their associated professional UMLS concepts, demonstrating close relationships between the two. The CUI in this table refers to the Concept Unique Identifier that the UMLS uses to identify its biomedicine concepts.

The MedlinePlus vocabulary was constructed to be the source of index terms for the MedlinePlus search engine (*Miller, Lacroix & Backus, 2000*). The NLM updates this resource yearly. In the 2018 UMLS version, there were 2112 professional concepts mapped to their laymen terms from the MedlinePlus topics. Due to the extensive human effort required, when we compared the 2018 UMLS version to the 2020 UMLS version, we found that only 28 new concepts had been mapped to their associated laymen terms. This slow rate of growth motivates the development of tools and algorithms to boost progress in mapping between professional and laymen.

Our research enriches laymen's vocabularies automatically based on healthcare text and seed terms from existing laymen vocabularies. Our system uses the Global Vector for Word Representations (GloVe) to build word embeddings to identify words similar to already existing laymen terms in a consumer health vocabulary. These potential matches are ranked by similarity and top matches are added to their associated medical concept. To improve the identification of new laymen terms, the GloVe results were enhanced by adding hyponyms, hypernyms, and synonyms from a well-known English ontology, WordNet. We make three main contributions:

- Developing an entirely automatic algorithm to enrich consumer health vocabularies automatically by identifying new laymen terms to be added entirely automatically.
- Developing an entirely automatic algorithm to add laymen terms to formal medical concepts that currently have no associated laymen terms.

- Improving GloVe algorithm results by using WordNet with a small, domain-specific corpora to build more accurate word embedding vectors.

Our work differs from others in that it is not restricted to a specific healthcare domain such as *Cancer*, *Diabetes*, or *Dermatology*. Moreover, our improvement is tied to enhancing the text and works with the unmodified GloVe algorithm. This allows for different word embedding algorithms to be applied. Furthermore, expanding small-size corpus with words from standard sources such as the WordNet eliminates the need to download large-size corpus, especially in a domain that is hard to find large text related to it.

## RELATED WORK

### Ontology creation

The past few years have witnessed an increased demand for ontologies in different domains (*Bautista-Zambrana, 2015*). According to *Gruber (1995)*, any ontology should comply with criteria such as clarity, coherence, and extensibility to be considered as a source of knowledge that can provide shared conceptualization (*Gruber, 1995*). However, building ontologies from scratch is immensely time-consuming and requires a lot of human effort (*Maedche & Staab, 2001*). Algorithms that can build an ontology automatically or semi-automatically can help reducing time and labor required to construct that ontology. *Zavitsanos et al. (2007)* presented an automatic method to build an ontology from scratch using text documents. They build that ontology using the Latent Dirichlet Allocation model and an existing ontology. *Kietz, Maedche & Volz (2000)* prototyped a company ontology semi-automatically with the help of general domain ontology and a domain-specific dictionary. They started with a general domain ontology called the GermanNet ontology. There are many other recent works presented to build ontologies from scratch, automatically or semi-automatically such as (*Hier & Brint, 2020*; *Yilahun, Imam & Hamdulla, 2020*; *Sanagavarapu, Iyer & Reddy, 2021*; *Mellal, Guerram & Bouhalassa, 2021*).

Ontologies should not be static. Rather, they should grow as their domains develop enriching existing ontologies with new terms and concepts. *Agirre et al. (2000)* used internet documents to enrich the concepts of WordNet ontology. They built their corpus by submitting the concept's senses along with their information to get the most relevant webpages. They used statistical approaches to rank the new terms (*Agirre et al., 2000*). A group at the University of Arkansas applied two approaches to enrich ontologies; (1) a lexical expansion approach using WordNet; and (2) a text mining approach. They projected concepts and their instances extracted from already existing ontology to the WordNet and selected the most similar sense using distance metrics (*Luong, Gauch & Wang, 2009*; *Luong, Gauch & Wang, 2012*; *Wang, Gauch & Luong, 2010*; *Luong, Gauch & Speretta, 2009*; *Luong, Gauch & Wang, 2009*; *Luong et al., 2009*; *Speretta & Gauch, 2008*). Recently, Ali and his team employed multilingual ontologies and documents to enrich not only domain-specific ontologies but also multilingual and multi-domain ontologies (*Ali et al., 2019*).

## Medical ontologies

The emphasis on developing an Electronic Health Record (EHR) for patients in the United States encouraged the development of medical ontologies to ensure interoperability between multiple medical information systems (*Rector, Qamar & Marley, 2009*; *Donnelly, 2006*). There are several healthcare vocabularies that provide human and machine-readable medical terminologies. The Systematized Nomenclature of Medicine Clinical Terms, SNOMED CT, is a comprehensive clinical ontology. It contains more than 300,000 professional medical concepts in multiple languages that has been adopted by many healthcare practitioners (*Donnelly, 2006*). Another professional vocabulary is the Royal Society of Chemistry's Name reaction Ontology (RXNO). The RXNO has over 500 reactions describing different chemical reactions that require organic compounds (*Schneider et al., 2016*). Recently, He and his team presented the coronavirus ontology with the purpose of providing machine-readable terms related to the coronavirus pandemic that occurs in 2020. This ontology includes all related coronavirus topics such as diagnosis, treatment, transmission, and prevention areas (*He et al., 2020*).

Medical ontologies, like all other ontologies, need to grow and adapt from time to time. *Zheng & Wang (2008)* prototyped the Gene Ontology Enrichment Analysis Software Tool (GOEST). It is a web-based tool that uses a list of genes from the Gene Ontology and enriches them using statistical methods. Recently, *Shanavas et al. (2020)* presented a method to enrich the UMLS concepts with related documents from a pool of professional healthcare documents. Their aim was to provide retrieval systems with more information about medical concepts.

### Consumer health vocabularies

Ontologies developed to organize professional vocabularies are of limited benefit in retrieval systems used by laypeople. Laymen usually use the lay language to express their healthcare concerns. Having a consumer health vocabulary can bridge the gap between the users' expression of their health questions and documents written using professional language. Consumer Health Vocabularies are particularly difficult to construct because they typically require knowledge of a specialized domain (medicine), a source of laymen's discussion about medicine, and finally a mechanism to map between the two. *Zeng et al. (2005)* detected and mapped a list of consumer-friendly display (CFD) names into their matched UMLS concepts. Their semi-automatic approach used a corpus collected from queries submitted to a MedlinePlus website. Their manual evaluation ended with mapping CFD names to about 1,000 concepts (*Zeng et al., 2005*; *Zeng & Tse, 2006*). Zeng's team continued working on that list of names to build what is called now the OAC CHV. In their last official update to this vocabulary, they were able to define associated laymen terms to about 56,000 UMLS medical concepts (*Doing-Harris & Zeng-Treitler, 2011*).

Several methods have been proposed to enrich such consumer vocabulary, such as *He et al. (2017)* who used a similarity-based technique to find a list of similar terms to a seed term collected from the OAC CHV. *Gu et al. (2019)* also tried to enriched the laymen vocabularies leveraging recent word embedding methods. A most recent work presented a method to enhance the consumer health vocabulary by associating laymen

terms with relations from the MeSH ontology (*Monselise et al., 2021*). Other recent work showed the variety of consumer vocabularies people use when writing their reviews (*Hovy, Melumad & Inman, 2021*). Previous research on enriching consumer health vocabularies were either semi-automatic or it did produce an automatic system accurate enough to be used in practice. Our automatic approach uses a recent word embedding algorithm, GloVe, which is further enhanced by incorporating a lexical ontology, WordNet. We work with gold standard datasets that are already listed on the biggest biomedicine ontology, the UMLS. This paper extends work we published in *Ibrahim et al. (2020)* by including additional datasets for evaluation and incorporating new approaches to improve the GloVe algorithm. These approaches leverage standard auxiliary resources to enrich the occurrence of laymen terms in the corpus. Although there are many large text corpora for general Natural Language Processing (NLP) research, there are far fewer resources specific to the healthcare domain. In order to extract information about domain-specific word usage by laymen for healthcare, we need to construct a domain-specific corpus of laymen's text. To our knowledge, we are the first to leverage MedlinePlus in order to automatically develop consumer health vocabularies.

### Finding synonyms to enrich laymen vocabularies

Our work focuses on finding new synonyms, words with the same meaning, to already existing laymen terms. Recent methods of finding synonyms are based on the idea that a word can be defined by its surroundings. Thus, words that appear in similar contexts are likely to be similar in meaning. To study words in text, they need to be represented in a way that allows for computational processing. Word vector representations are a popular technique that represents each word using a vector of feature weights learned from training texts. In general, there are two main vector-learning models. The first models incorporate global matrix factorization whereas the second models focus on local context windows. The global matrix factorization models generally begin by building a corpus-wide co-occurrence matrix and then they apply dimensionality reduction. An early example of this type of model is Latent Semantic Analysis (LSA) (*Deerwester et al., 1990*) and Latent Dirichlet Allocation (LDA) (*Blei, Ng & Jordan, 2003*). The context-window models are based on the idea that a word can be defined by its surroundings. An example of such models is the skip-gram model (*Mikolov, Yih & Zweig, 2013*) proposed by Mikolov in 2013 and the model proposed by *Gauch, Wang & Rachakonda (1999)*. Word2Vec (*Mikolov et al., 2013b*), FastText (*Bojanowski et al., 2017*), GloVe (*Pennington, Socher & Manning, 2014*) and WOVe (*Ibrahim et al., 2021*) are all examples of vector learning methods that have been shown to be superior to traditional NLP methods in different text mining applications (*Gu et al., 2019*; *Hasan & Farri, 2019*). Some of these techniques have been applied in the medical field to build medical ontologies, such as (*Minarro-Giménez, Marin-Alonso & Samwald, 2014*; *Hughes et al., 2017*; *De Vine et al., 2014*; *Minarro-Giménez, Marín-Alonso & Samwald, 2015*; *Wang, Cao & Zhou, 2015*).

Our work focuses on the word similarity task, or specifically word synonyms task. In order to find these synonyms, we leveraged the GloVe algorithm. This algorithm has outperformed many vector learning techniques in the task of finding word similarity

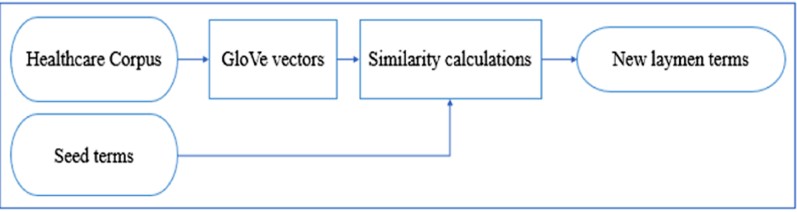

**Figure 1 Methodology of finding new laymen terms.**

(*Pennington, Socher & Manning, 2014*). It combines the advantages of two vector learning techniques: global matrix factorization methods and local context window methods (*Pennington, Socher & Manning, 2014*). This algorithm has many applications in different fields such as text similarity (*Kenter & De Rijke, 2015*), node representations (*Brochier, Guille & Velcin, 2019*), emotion detection (*George, Barathi Ganesh & Soman, 2018*) and many others. This algorithm found its way in many biomedicine such as finding semantic similarity (*Muneeb, Sahu & Anand, 2015*), extracting Adverse Drug Reactions (ADR) (*Lin et al., 2015*), and analyzing protein sequences (*George et al., 2019*).

GloVe is generally used with very large corpora, *e.g.*, a 2010 Wikipedia corpus (1 billion tokens), a 2014 Wikipedia corpus (1.6 billion tokens), and Gigaword 5 (4.3 billion tokens) (*Pennington, Socher & Manning, 2014*). In comparison, our corpus is specialized and much smaller, approximately 1,365,000 tokens. To compensate for the relative lack of training text, we incorporate an auxiliary source of vocabulary, WordNet. The WordNet is a machine-readable English ontology proposed by Professor George A. at the Princeton University. The most recent version has about 118,000 synsets (synonyms) of different word categories such as noun, verb, adjective, and adverb. For every synset, WordNet provides a short definition and sometimes an example sentence. It also includes a network of relations between its synsets. The synonyms, antonyms, hyponyms, hypernyms, meronymy's and some others are all semantic relations that WordNet provides (*Miller, 1995*). WordNet has been used in many fields to help enrich ontologies in different domains such as (*Espinoza, Gómez-Pérez & Mena, 2008*; *Navigli & Velardi, 2006*; *Warin, Oxhammar & Volk, 2005*).

## METHODOLOGY

Figure 1 illustrates the main steps of our algorithm. Our method starts with a corpus collected from a healthcare social media platform to be used as the source of the new laymen terms. Using this corpus, the GloVe algorithm builds word embeddings. For every UMLS medical concept, there is a list of its associated laymen terms from which we select a seed terms for the concept. Using the GloVe vectors and a similarity metric, we identify the most similar words to these seed term and choose the top-ranked candidate as a new layman term. The next sections explain the methodology steps in detail.

### Healthcare corpus

To find new laymen terms, we need text documents that can be used as a source of new laymen terms. Because of the specialized nature of medical terminology, we need

domain-specific text related to the field of healthcare. *MedHelp.org* is a healthcare social media platform that provides a question/answer for people who share their healthcare issues. In this platform, the lay language is used more than formal medical terminology. Instead of writing a short query on the internet that may not retrieve what a user is looking for, whole sentences and paragraphs can be posted on such media (*Kilicoglu et al., 2018*) and other members of the community can provide answers. People might use sentences such as "I can't fall asleep all night" to refer to the medical term "insomnia" and "head spinning a little" to refer to "dizziness" (*Tutubalina et al., 2018*). Such social media can be an excellent source from which to extract new laymen terms.

## Seed term list

Our task is to enrich formal medical concepts that already have associated laymen terms by identifying additional related layperson terms. These associated terms are used as seed terms that the system uses to find synonyms and then these synonyms are added to that medical concept. To do so, we need an existing ontology of medical concepts with associated laymen vocabulary. For our experiment, we used two sources of laymen terms: OAC CHV (*Doing-Harris & Zeng-Treitler, 2011*), and the MedlinePlus consumer vocabulary (*Miller, Lacroix & Backus, 2000*). The OAC CHV covers about 56,000 concepts of the UMLS, and the MedlinePlus mapped to about 2,000 UMLS concepts.

## Synonym identification algorithms

This paper reports on the results of applying several algorithms to automatically identify synonyms of the seed terms to add to existing laymen's medical concepts. The algorithms we evaluated are described in the next section.

### Global vectors for word representations (GloVe)

Our baseline approach uses an unmodified version of GloVe to find the new laymen terms trained only on our unmodified corpus. As reported in *Pennington, Socher & Manning (2014)*, GloVe starts collecting word contexts using its global word to word co-occurrence matrix. This matrix is a very large and very sparse matrix that is built during a onetime pass over the whole corpus. Given a word to process, *i.e.*, the *pivot* word, GloVe counts co-occurrences of words around the pivot word within a window of a given size. As the windows shift over the corpus, the pivot words and contexts around them continually shift until the matrix is complete. GloVe builds word vectors for each word that summarize the contexts in which that word was found. Because the co-occurrence matrix is very sparse. GloVe uses the log bilinear regression model to build reduce the dimensionality of the co-occurrence matrix. This model also optimizes word vectors by tuning its weights and reducing errors iteratively until finding the best word representations. By comparing the seed terms words vectors with all other word vectors using the cosine similarity measure, highly similar words, *i.e.*, potential new laymen terms, can be located. The unmodified GloVe algorithm is our baseline to compare with the GloVe improvement methods.

**Peer**J Computer Science

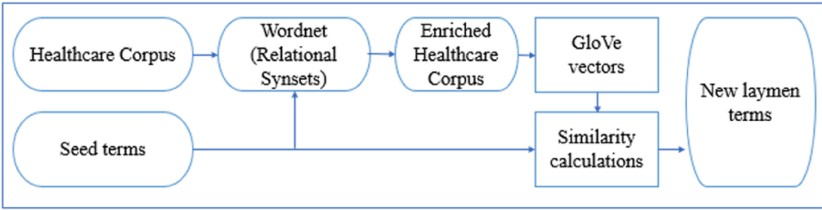

**Figure 2 Methodology of improved GloVe with WordNet corpus enhancement.**

### GloVe with WordNet

Word embedding algorithms usually use a very large corpus to build their word representations, *e.g.*, 6B words of Google News corpus are used to train the word2vec vectors (*Mikolov et al., 2013a*, *2013b*). In the case of a narrow domain such as healthcare, it is hard to find or build an immense corpus, increasing the sparsity of the co-occurrence matrix and impacting the accuracy of the resulting word vectors. Thus, one of our goals is to investigate the ability of an external ontology to increase the accuracy of word embeddings for smaller corpora. In particular, we present methods to exploit a standard English ontology, WordNet, to enhance GloVe's accuracy on a healthcare domain corpus. WordNet provides a network of relations between its relational synsets such as, synonyms, antonyms, hyponyms, hypernyms, meronymy's and some other relations.

In our research, we investigate using the synonym, hyponym, and hypernym relations to augment our corpus prior to running GloVe. We only expand the seed terms in the corpus with their relational synsets. For each seed term, we located the relational synset of interest, *e.g.*, hyponyms we sort them by similarity to the seed term using the *Resnik (1995)* similarity measurement. Then, we limit the list to not exceed the 10th most similar synsets. We split them evenly into two subsets of roughly equal total similarity using a round-robin algorithm. We then expand the corpus by adding the first subset of relational synset words to the corpus prior to each seed term occurrence and the second subset after each seed term occurrence. One of issues that arises when using WordNet is the polysemy of its synsets. Many words are ambiguous and thus map to synsets with different meanings adding noise to the expanded context vectors. By limiting the number of words used from WordNet and incorporating words from the context around the seed term, the effect of noise on GloVe's model is decreased. In future, we could explore using the context words around the seed terms to identify the best synsets to use for expansion. Figure 2 shows the methodology of our system with the WordNet ontology.

Expressing the WordNet method, let $S = \{s_1, s_2, s_3, \ldots, s_n\}$ be a set of $n$ seed terms. Let $T =$ "$w_1\ w_2\ w_3 \ldots w_k$" be a text of words in the training corpus. Let $X = \{x_1, x_2, x, \ldots, x_z\}$ be a set of relational synset terms for the seed term $s_i$, where $i = 0,1,2,\ldots,n$. These relational synsets are sorted according to their degree of similarity to $s_i$ using the Resnik similarity measurement (*Resnik, 1995*). $X$ is sorted according to Resnik score and divided into two sets $X_1$ and $X_2$. Each set goes to one side of $s_i$. Now, let

$s_i = w_{j+2}$ in T, where $j = 0,1,2,3,\ldots,k$. Then, the new text $\hat{T}$ after adding the relational synsets will look like this:

$$\hat{T} = \text{``}w_j w_{j+1} X_1 w_{j+2} X_2 \ldots w_{j+k}\text{''}$$

Further, consider the effect of T-hat on the GloVe cooccurence vectors. Assume that $s_i$ has the vector $\overrightarrow{V_{s_i}}$. Assume that $X$ has the vector $\overrightarrow{V_X}$. After expanding the training corpus with the relational synsets, the new vector $\overrightarrow{V_{s_i}}$ will equal:

$$\overrightarrow{\check{V}_{s_i}} = \overrightarrow{V_{s_i}} + \overrightarrow{V_X} \tag{1}$$

The co-occurrence weights of relational synsets that are already in the corpus will be increased incrementally in the vector, while those that are new to the corpus will expand the vector and their co-occurrence weight will be calculated according to the co-occurrence with the seed term. The following sections outline the WordNet approach above with the three types of relational synsets we used: synonyms, hyponyms, and hypernyms.

*GloVe WordNet Synonyms (GloVeSyno)*

Synonyms are any words that share the same meaning. For example, the words *auto*, *machine*, and *automobile* are all synonyms of the word *car*. Having synonyms around a seed term adds more information about that seed term and help building more accurate seed term vectors. When a seed term found in the training corpus, WordNet provides a list of its synonyms. These synonyms are sorted according to their degree of similarity to the seed term. After that, the synonyms are divided into two lists and each list go to one side of the seed term. Here is an example that demonstrate this process. Let $T = \text{``}I\ had\ a\ headache\text{''}$ be a text in the training corpus. $T$ has the seed term $s = headache$. The WordNet synonyms of this seed term are {*concern, worry, vexation, cephalalgia*}. Sorting this set according to their degree of similarity results the following set: {*worry, cephalalgia, concern, vexation*}. This set is divided in to two sets {*worry, cephalalgia*} and {*concern, vexation*} and added to the left and right of the $s$ in $T$. So, the $\hat{T}$ equals:

$$\hat{T} = \text{``}I\ had\ a\ worry\ cephalalgia\ headache\ concern\ vexation\text{''}$$

Assume that the vector of the seed term $s$, $\overrightarrow{V_s}$, before expanding the training corpus looks like this:

| $\overrightarrow{V_s}$ | dizzy | pain | I | had | a | for | worry | please | sleep |
|---|---|---|---|---|---|---|---|---|---|
| | 5 | 0 | 5 | 10 | 1 | 0 | 15 | 0 | 50 |

The $\overrightarrow{\check{V}_s}$ for the seed term after expanding the training corpus with the WordNet synonyms will be expanded to have the new words and updated the occurrence of the already in corpus words. Here is how the $\overrightarrow{\check{V}_s}$ looks like:

| $\overrightarrow{\check{V}_s}$ | Cephalalgia | dizzy | pain | I | had | a | for | concern | worry | please | vexation | sleep |
|---|---|---|---|---|---|---|---|---|---|---|---|---|
| | 1 | 5 | 0 | 5 | 10 | 1 | 0 | 1 | 16 | 0 | 1 | 50 |

We can see from the $\overrightarrow{\check{V}_s}$ that the words that are new to the corpus vocabulary expanded the vector and their weights are calculated according to their co-occurrence with the

seed term, while the words that are already in the vector, such as *worry*, their weights increased incrementally.

*Glove WordNet Hyponyms (GloVeHypo)*

Hyponyms are those words with more specific meaning, *e.g.*, *Jeep* is a hyponym of *car*. The idea here is to find more specific names of a seed term and add them to the context of that seed term. to explain this method, we use the same example we used in the previous section. The hyponyms of the seed term *headache* that the WordNet provides are {*dead_weight, burden, fardel, imposition, bugaboo, pill, business*}. Sorting these hypos according to their degree of similarity to the seed term results the set {*dead_weight, burden, fardel, bugaboo, imposition, business, pill*}. This list is divided into two sets and each set go to one of the seed term's sides. The rest process is the same as the GloVeSyno method.

*GloVe WordNet Hypernyms (GloVeHyper)*

Hypernyms are the antonyms of hyponyms. Hypernyms are those words with more general meaning, *e.g.*, *car* is a hypernym of *Jeep*. The idea here is to surround a seed term with more general information that represents its ontology. Having this information leads to more descriptive vector that represent that seed term. An example of a seed term hypernyms is the hypernyms of the seed term *headache*, which are {*entity, stimulation, negative_stimulus, information, cognition, psychological_feature, abstraction*}. We can see that these hypernyms are broader than the seed term *headache*. We use the same steps for this relational synset as in the GloVeSyno method by sorting, dividing, and distributing these hypernyms around the seed term in the corpus. After that GloVe builds its co-occurrence matrix from the expanded corpus and builds its word vectors that are used to extract the terms most similar terms to the seed terms from the corpus.

## Similarity measurement

We use cosine similarity between word vectors to find the terms most similar word to the seed term. This metric is widely used for vector comparisons in textual tasks such as (*Salton, Wong & Yang, 1975*; *Habibi & Cahyo, 2020*; *Park, Hong & Kim, 2020*; *Zheng et al., 2020*). Because it focuses on the angle between vectors rather than distances between endpoints, cosine similarity can handle vector divergence in large-size text documents better than Euclidean distance (*Prabhakaran, 2018*). As such, it is one of the most commonly used similarity based metrics (*Polamuri, 2015*; *Gupta, 2019*; *Lüthe, 2019*). Cosine similarity (Eq. 2) produces a score between 0 and 1, and the higher the score between two vectors, the more similar they are (*Singhal, 2001*).

$$\text{cos\_sim}(v_1, v_2) = \frac{\vec{v_1} \cdot \vec{v_2}}{|\vec{v_1}||\vec{v_2}|}, \tag{2}$$

where $\vec{v_1}$ is a vector of a seed term in the seed term list, and $\vec{v_2}$ is a vector of a word in the corpus that GloVe model built. We consider the terms in the list *candidate terms* for inclusion. The top $n$ candidate terms are the new laymen terms that we add to the UMLS concept.

**Table 2  MedHelp.org community corpus statistics.**

| No. | Community | Posts | Tokens |
| --- | --- | --- | --- |
| 1. | Addiction | 82,488 | 32,871,561 |
| 2. | Pregnancy | 308,677 | 33,989,647 |
| 3. | Hepatitis-C | 46,894 | 21,142,999 |
| 4. | Neurology | 62,061 | 9,394,044 |
| 5. | Dermatology | 67,109 | 8,615,484 |
| 6. | STDs/STIs | 59,774 | 7,275,289 |
| 8. | Gastroenterology | 43,394 | 6,322,356 |
| 9. | Women health | 66,336 | 5,871,323 |
| 10. | Heart Disease | 33,442 | 5,735,739 |
| 11. | Eye Care | 31,283 | 4,281,328 |
| Total | | 801,458 | 135,499,770 |

# EVALUATION

## Corpus

*MedHelp.org* has many communities that discuss different healthcare issues such as Diabetes Mellitus, Heart Diseases, Ear and Eye care, and many others. To select the communities to include in our dataset, we did an informal experiment to find the occurrences of laymen terms from the OAC CHV vocabulary on *MedHelp.org*. We found that the highest density of these OAC CHV terms occur in communities such as Pregnancy, Women's Health, Neurology, Addiction, Hepatitis-C, Heart Disease, Gastroenterology, Dermatology, and Sexually Transmitted Diseases and Infections (STDs/STIs) communities. We thus chose these nine communities for our testbed and downloaded all the user-posted questions and their answers from *MedHelp.org* from WHEN to April 20, 2019. The resulting corpus is roughly 1.3 Gb and contains approximately 135,000,000 tokens. Table 2 shows the downloaded communities with their statistics.

We removed all stopwords, numbers, and punctuations from this corpus. We also removed corpus-specific stopwords such as test, doctor, symptom, and physician. Within our domain-specific corpus, these ubiquitous words have little information content. Finally, we stemmed the text using the Snowball stemmer (*Porter, 2001*), and removed any word less than 3 characters long. The final corpus size was ~900 mb. This corpus is available for download from *Gauch et al. (2020)*.

## Seed terms

We built the seed term list from the OAC CHV and MedlinePlus vocabularies, choosing seed terms with a unigram form, such as flu, fever, fatigue, and swelling. There are two reason that led us to choose only unigrams for our work. First, GloVe embeddings handle only single word vectors, and second the existing laymen vocabularies are rich with such seed terms. We also chose the professional medical concepts that had a unigram form, then we pick its unigram associated laymen terms. In many cases, the medical concept on these two vocabularies has associate laymen terms that have same names as the

**Table 3  UMLS concepts with their seed terms from the MedlinePlus dataset.**

| CUI | Medical concept | Associated Laymen terms | | |
|-----|-----------------|-------------------------|-----|-----|
| C0043246 | laceration | lacer | torn | tear |
| C0015672 | fatigue | weariness | tired | fatigued |
| C0021400 | influenza | flu | influenza | grippe |

concept's name except different morphological forms, such the plural 's', uppercase/ lowercase of letters, punctuations, or numbers. We treated these cases and removed any common medical words. After that, we stemmed the terms and listed only the unique terms. For example, the medical concept *Tiredness* has the laymen terms *fatigue*, *fatigues*, *fatigued* and *fatiguing*. After stemming, only the term 'fatigu' was kept. To focus on terms for which sufficient contextual data was available, we kept only these laymen terms that occur in the corpus more than 100 times.

To validate our system, we need at least two terms for each professional medical concept, one term is used as the seed laymen term and one to be used as the target term for evaluation. Thus, we kept only those medical concepts that have at least two related terms. From the two vocabularies, we were able to create an OAC CHV ground truth dataset of size 944 medical concepts with 2,103 seed terms and a MedlinePlus ground truth dataset of size 101 medical concepts with 227 seed terms. Table 3 shows an example of some UMLS medical concepts and their seed terms from the MedlinePlus dataset. When we run our experiments, we select one seed term at random from each of the 944 concepts in the ground truth dataset for testing. We evaluate each algorithm's ability to identify that seed term's ground truth synonyms using the metrics described in the next section.

The OAC CHV dataset is nine times bigger than the MedlinePlus dataset (see Fig. 3A); the OAC CHV vocabulary covers 56,000 of the UMLS concepts whereas the MedlinePlus covers only 2,112 UMLS concepts. Although it is smaller, MedlinePlus represents the future of laymen terms because the NLM updates this resource annually. In contrast, the last update to the OAC CHV was in 2011. Figure 3B shows that 37% of the 101 concepts in MedlinePlus also appear in the OAC CHV dataset and share the same concepts and laymen terms. This indicates that the OAC CHV is still a good source of laymen terms.

## Baselines and metrics

We consider the basic GloVe results as the baseline for comparison with the WordNet expansion algorithms. First, we tune the baseline to the best setting. We then compare the results when we use those settings with our WordNet-expanded corpora. We evaluate our approach using precision (P), recall (R), and F-score (F), which is the harmonic mean of the previous two (*Powers, 2011*). We also include the number of concepts (NumCon) that the system could find one or more of its seed terms. Moreover, we include the Mean Reciprocal Rank (MRR) (*Voorhees & Harman, 2000*) that measures the rank of the first most similar candidate term in the candidate list. It has a value between 0 and 1, and the closer the MRR to 1, the closer the candidate term position in the candidate list.

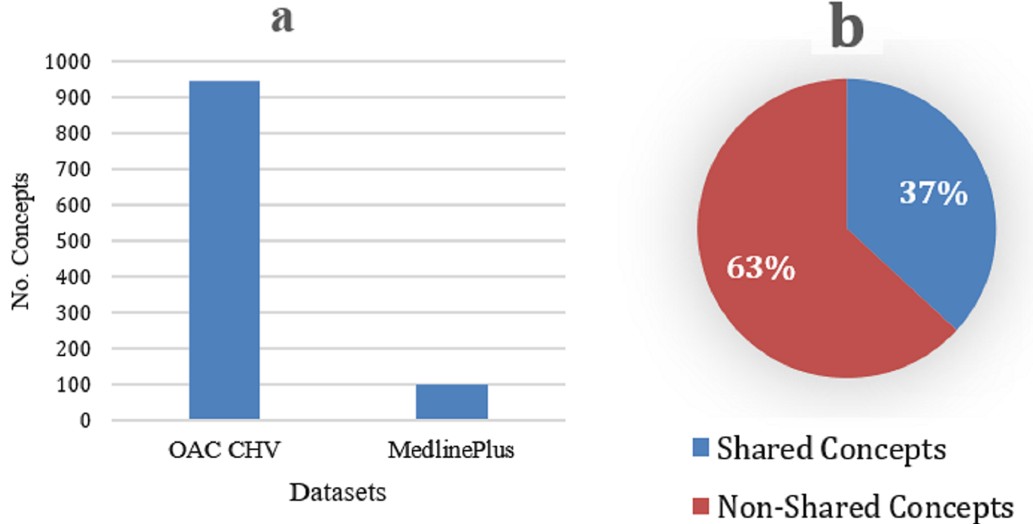

**Figure 3** (A) Size of the OAC CHV dataset to the MedlinePlus dataset. (B) Shared concepts and their laymen terms between the MedlinePlus and OAC CHV datasets.

Based on a set of medical concepts for which we have a seed term and at least one synonym in the ground truth data set, we can measure the precision, recall, and F-Score metrics according to two criteria: (1) the number of concepts for which the system was able to find at least one synonym; and (2) the total number of synonyms for seed terms the system was able to find across all concepts. We call the metrics used to measure these two criteria the macro and micro average metrics, respectively. The macro average measures the number of the concepts for which the algorithm found a match to the ground truth dataset while the micro average measures the number of new terms found. The micro and macro precision, recall, and F-score are computed according to these equations:

$$P_{micro} = \frac{\text{\# of true synonyms in the candidate lists}}{\text{total \# of terms in the candidate lists}}, \tag{3}$$

$$R_{micro} = \frac{\text{\# of true synonyms in the candidate lists}}{\text{total \# of synonms in the ground truth dataset}}, \tag{4}$$

$$P_{macro} = \frac{\text{\# of concepts whose candidate list contains a true synonym}}{\text{total \# of concepts}}, \tag{5}$$

$$R_{macro} = \frac{\text{\# of concepts whose candidate list contains a true synonym}}{\text{total \# of concepts in the ground truth dataset}}, \tag{6}$$

$$F-score = 2 * \frac{Precision * Recall}{Precision + Recall}, \tag{7}$$

We illustrate these measurements in the following example. Suppose we have a ground truth dataset of size 25 concepts, and every concept has four synonyms terms. For every concept, a random synonym term selected to be a seed term. The remaining 75 synonyms will be used for evaluation. Suppose the algorithm retrieves five candidate terms for each seed term and it is able to generate results for 20 of the seed terms, creating

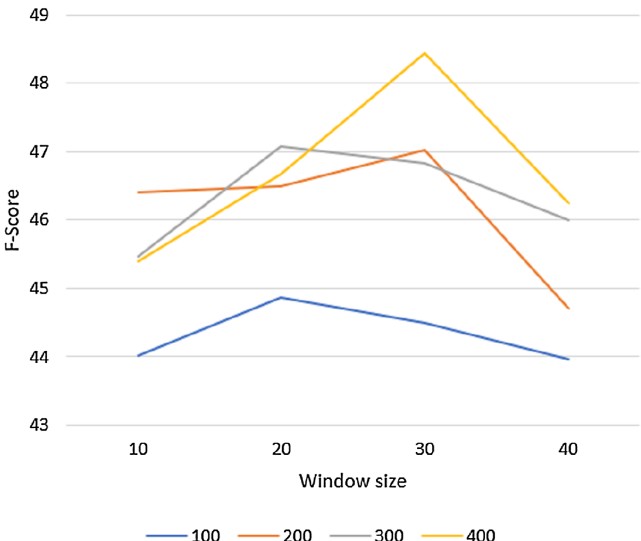

**Figure 4 The Macro F-Score for the GloVe algorithm with different vector and window sizes.**

20 candidate term lists. That makes 100 candidate terms in total. Assume that only 15 out of the 20 candidate lists contain a true synonym, and each list of those 15 lists includes two true synonyms. Thus, this algorithm extracted 30 true laymen terms. Having all this information, then the $P_{micro} = 30/100$, $R_{micro} = 30/75$, $P_{macro} = 15/20$, and $R_{macro} = 15/25$.

## RESULTS

### Experiment 1: tuning GloVe to the best setting

To tune the GloVe algorithm to its best setting, we varied GloVe's parameters on an unmodified corpus and we used the larger of our two datasets, the OAC CHV for testing. The GloVe algorithm has many hyperparameters, but the vector size and the window size parameters have the biggest effect on the results. We evaluated GloVe using the 944 concepts in this dataset on different vector sizes (*100, 200, 300, 400*), varying the window size (*10, 20, 30, 40*) for each vector size. We set the candidate list size to $n = 10$. Figure 4 shows the macro F-score results of the GloVe algorithm according to these different vector and window sizes. In general, the F-score results declined with any window size greater than 30.

Table 4 reports the micro accuracy for GloVe results, and the focus in this table is on the micro-precision that we have highlighted in bold. We can see that the micro precision is very low due to the size of the candidate lists created. In particular, we are testing with 944 concept seed terms, and the size of the candidate list is set to 10, so we generate $944 \times 10 = 9,440$ candidate terms. However, there are only 2,103 truth synonyms, to the micro-averages are guaranteed to be quite low. To compensate, we need to determine a good size for the candidate list that balances recall and precision. This is discussed further in Experiment 3.

The highest F-score was reported with a vector of size 400 and window of size 30. Thus, we used these settings for all following experiments.

**Table 4 The micro-precision of GloVe.**

| Vector size | NumCon | Micro | | |
|---|---|---|---|---|
| | | Precision | Recall | F-score |
| 100 | 420 | **4.78** | 38.91 | 8.51 |
| 200 | 444 | **5.07** | 41.33 | 9 |
| 300 | 442 | **5.16** | 42.02 | 9.19 |
| 400 | 457 | 5.28 | 42.97 | 9.41 |

Note:
The micro-precision is highlighted in bold.

**Table 5 Evaluation of the basic GloVe, GloVeSyno, GloVeHypo, and GloVeHyper algorithms over the OAC CHV and MedlinePlus datasets.**

| | NumCon | Macro | | | MRR |
|---|---|---|---|---|---|
| | | Precision | Recall | F-score | |
| OAC CHV | | | | | |
| Basic GloVe | 457 | 48.46 | 48.41 | 48.44 | 0.29 |
| GloVeSyno | **546** | **57.9** | **57.84** | **57.87** | **0.35** |
| GloVeHypo | 280 | 29.69 | 29.66 | 29.68 | 0.33 |
| GloVeHyper | 433 | 45.92 | 45.87 | 45.89 | 0.35 |
| MedlinePlus | | | | | |
| Basic GloVe | 48 | 51.06 | 47.52 | 49.23 | 0.38 |
| GloVeSyno | **63** | **66.32** | **62.38** | **64.29** | **0.36** |
| GloVeHypo | 32 | 33.33 | 31.68 | 32.49 | 0.37 |
| GloVeHyper | 35 | 37.23 | 34.65 | 35.9 | 0.35 |

Note:
The highest results reported by the algorithms over the two ground truth datasets are highlighted in bold.

## Experiment 2: GloVe with WordNet

Using the best GloVe setting reported in the previous experiment, we next evaluate the GloVeSyno, GloVeHypo, and GloVeHyper algorithms to determine whether or not they can improve on basic GloVe's ability to find layman terms. After processing the corpus, we expand our laymen's corpus with synonyms, hyponyms, and hypernyms from WordNet, respectively. These are then input to the GloVe algorithm using 400 for the vector size and 30 for the window size. Table 5 shows a comparison between the results of these WordNet algorithms, and our baseline GloVe for the OAC CHV and MedlinePlus datasets. The evaluation was done using a candidate list of size $n = 10$. We report here the macro accuracy of the system for the three algorithms which is based on the number of concepts for which a ground truth result was found. The bold results in Table 5 shows the highest results reported by the algorithms over the two ground truth datasets.

We can see from Table 5 that GloVeSyno outperformed the other algorithms. It was able to enrich synonyms to 57% (546) of the medical concepts listed in the OAC CHV dataset and more than 62% (63) of the concepts in the MedlinePlus dataset. Table 6 presents the algorithms' performance averaged over the two datasets. Best results have

**Table 6 The average results of the basic GloVe, GloVeSyno, GloVeHypo, and GloVeHyper algorithms over the OAC CHV and MedlinePlus datasets.**

| Algorithm | NumCon | Macro | | | MRR | F-score Rel-Improv. |
|---|---|---|---|---|---|---|
| | | Precision | Recall | F-score | | |
| Basic GloVe | 252.5 | 49.76 | 47.965 | 48.835 | 0.335 | |
| GloVeSyno | **304.5** | **62.11** | **60.11** | **61.08** | **0.355** | 25% |
| GloVeHypo | 156 | 31.51 | 30.67 | 31.085 | 0.350 | −36% |
| GloVeHyper | 234 | 41.575 | 40.26 | 40.895 | 0.350 | −16% |

**Note:**
Best results have been highlighted with bold.

been highlighted with bold. On average, the GloVeSyno algorithm produced an F-score relative improvement of 25% comparing to the basic GloVe. Moreover, the GloVeSyno reported the highest MRR over all the other algorithms, which shows that the first most similar candidate term to the seed term fell approximately in the 2nd position of the candidate list. Furthermore, the GloVeSyno showed a high statistical significance over the two ground truth datasets with $P < 0.001$.

The GloVeHypo and GloVeHyper results were not good comparing to the other algorithms. The reason is that the hyponyms provide a very specific layman term synsets. For example, the hyponyms of the laymen term *edema* are *angioedema*, *atrophedema*, *giant hives*, *periodic edema*, *Quincke'*, *papilledema*, and *anasarca*. Such hypos are specific names of the laymen term *edema*, and they might not be listed in ground truth datasets. We believe that the GloVeHypo algorithm results are promising, but a more generalized and bigger size ground truth dataset is required to prove that.

On the other hand, the GloVeHyper algorithm was not good comparing to the basic GloVe algorithm. However, it is better than the GloVeHypo algorithm. The reason that this algorithm did not get a good result is because the degree of abstraction that the hypernym relations provide. For example, the hypernym *contagious_disease* represents many laymen terms, such as *flu*, *rubeola*, and *scarlatina*. Having such hypernym in the context of a layman term did not lead to good results. The hypernym *contagious_disease* is very general relation that can represent different kind of diseases.

To illustrate the effectiveness of the GloVeSyno algorithm, we show a seed term the candidate synonyms for a selection of concepts in Table 7. Although only 14 true synonyms from 7 concepts were found, we note that many of the other candidate synonyms seem to be good matches even though they do not appear in the official vocabulary. These results are promising and could be used to enrich medical concepts with missing laymen terms. They could also be used by healthcare retrieval systems to direct laypersons to the correct healthcare topic.

On average, the GloVeSyno algorithm outperformed all the others, producing an F-score relative improvement of 25% compared to basic GloVe. The results were statistically significant ($P < 0.001$). Additionally, this algorithm found many potentially relevant laymen terms that were not already in the ground truth.

**Table 7 Sample of the GloVeSyno output (seeds stemmed).**

| CUI | Seed Term | Candidate Synonyms | | | | | | |
|---|---|---|---|---|---|---|---|---|
| C0015967 | feverish | febric | **febril** | **pyrexia** | **fever** | chili_pepp | chilli | influenza |
| C0020505 | overeat | gormand | pig_out | ingurgit | gormandis | scarf_out | overindulg | gourmand |
| C0013604 | edema | **oedema** | **hydrop** | **dropsi** | swell | puffi | ascit | crestless |
| C0039070 | syncop | **swoon** | deliquium | **faint** | vasovag | neurocardi | dizzi | lighthead |
| C0015726 | fear | **fright** | **afraid** | **scare** | terrifi | *scari* | panic | anxieti |
| C0014544 | seizur | rictus | **seiz** | raptus | prehend | shanghaier | seizer | clutch |
| C0036916 | stds | **std** | gonorrhea | encount | chlamydia | hiv | herp | syphili |

**Note:**

The candidate synonyms that appear in the ground truth list of synonyms are highlighted with bold.

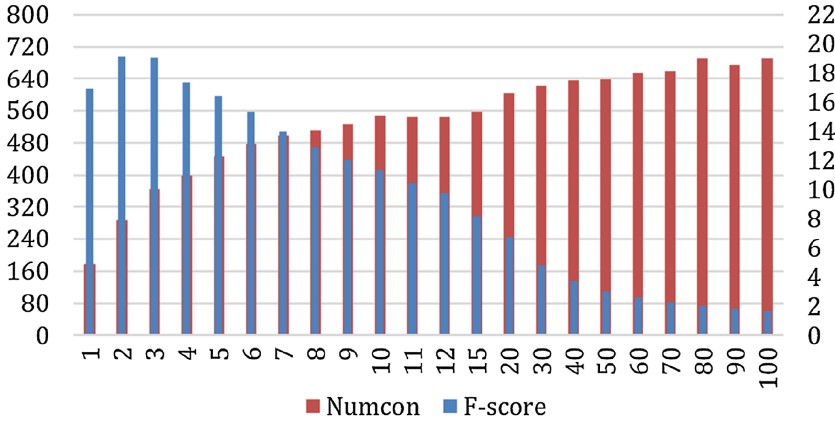

**Figure 5 Micro F-Score and the number of concepts for the GloVeSyno algorithm over the OAC CHV dataset.**          

We further examined the effect of the WordNet terms on the set of candidate terms extracted from the corpus. With GloVeSyno, 60% of the true positives appeared in the WordNet synsets used for corpus expansion versus 41% with unmodified GloVe. Thus, even after expanding with WordNet, 40% of the true positives appeared in the corpus and not in the nearest WordNet synsets, indicating that the external lexicon and GloVe's word vectors find complementary sets of synonyms.

## Experiment 3: improving the GloVeSyno micro accuracy

From our previous experiment, we conclude that the GloVeSyno algorithm was the most effective. However, we next explore it in more detail to see if we can improve its accuracy by selecting an appropriate number of candidate synonyms from the candidate lists. We report evaluation results according to the ground truth datasets, OAC CHV and MedlinePlus. We varied the number of synonyms selected from the candidate lists $n = 1$ to $n = 100$ and measured the micro recall, precision, and F-score. Figure 5 shows the F-score results and the number of concepts for which at least one true synonym was extracted. This figure reports the results of the GloVeSyno algorithm over the OAC CHV dataset. The F-score is maximized with $n = 3$ with an F-score of 19.06% and 365 out of 944

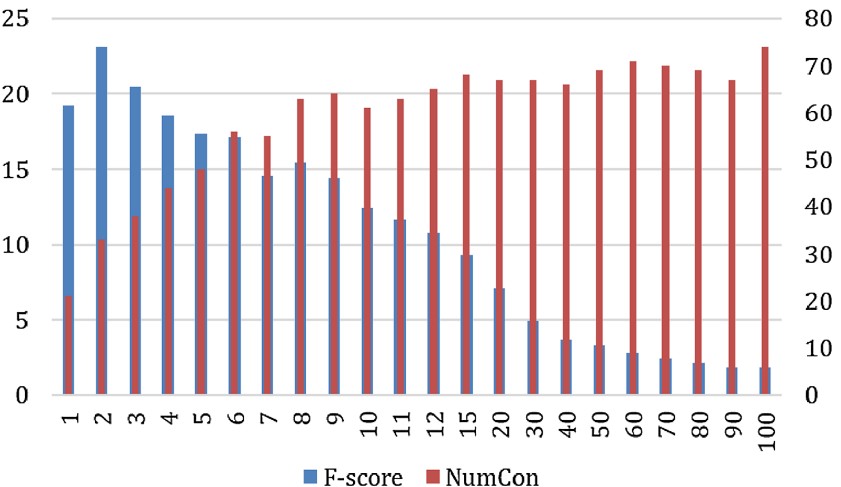

**Figure 6 Micro F-Score and the number of concepts for the GloVeSyno algorithm over the MedlinePlus dataset.**

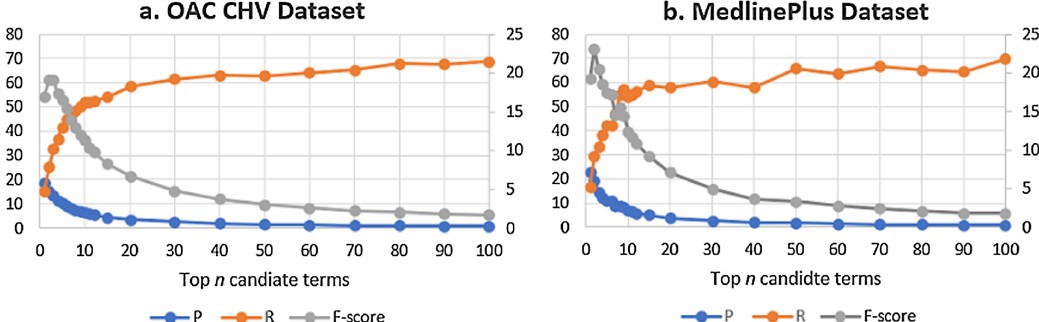

**Figure 7 (A & B) F-Score results over the precision and recall for the GloVeSyno algorithm over the OAC CHV and MedlinePlus datasets.**

concepts enriched. After that, it starts to decline quickly and at $n = 20$ the F-score is only 6.75% which further declines to 1.7% at $n = 100$. We note that the number of concepts affected rose quickly until $n = 7$, but then grows more slowly. The best results are with $n = 2$ with an F-score of 19.11%. At this setting, 287 of the 944 concepts are enriched with a micro-precision of 15.43% and recall of 25.11%.

The evaluation results over the MedlinePlus dataset looks the same as the results reported for the OAC CHV dataset (See Fig. 6). The F-score was at its highest score at $n = 2$ with an F-score of 23.12% and 33 out of 101 concepts enriched. The F-score decreased quickly at $n = 30$ and was at its lowest score at $n = 100$ with an F-score of 1.81%. The number of enriched concepts grew quickly until $n = 6$ and stabilized after $n = 9$ between 64 and 74 enriched concepts.

Over the two datasets, the best results are with $n = 2$. Figure 7 shows the F-score over the Precision and recall for the two datasets. Despite the difference in the number of concepts between the two ground truth datasets, the results show that the F-score is the best at $n = 2$. The figure shows that the behaviors of the GloVeSyno over the two datasets are almost the same over different candidate list settings.

Based on these results, we conclude that the best performance for automatically enriching a laymen vocabulary with terms suggested by GloVeSyno be achieved by adding the top two results.

## CONCLUSION AND FUTURE WORK

This paper presents an automatic approach to enrich consumer health vocabularies using GloVe word embeddings and an auxiliary lexical source, WordNet. Our approach was evaluated used a healthcare text downloaded from *MedHelp.org*, a healthcare social media platform using two standard laymen vocabularies, OAC CHV, and MedlinePlus. We used the WordNet ontology to expand the healthcare corpus by including synonyms, hyponyms, and hypernyms for each layman term occurrence in the corpus. Given a seed term selected from a concept in the ontology, we measured our algorithms' ability to automatically extract synonyms for those terms that appeared in the ground truth concept. We found that GloVeSyno and GloVeHypo both outperformed GloVe on the unmodified corpus, however including hypernyms actually degraded performance. GloVeSyno was the best performing algorithm with a relative improvement of 25% in the F-score versus the basic GloVe algorithm. Furthermore, the GloVeSyno showed a high statistical significance over the two ground truth datasets with $P < 0.001$.

The results of the system were in general promising and can be applied not only to enrich laymen vocabularies for medicine but any ontology for a domain, given an appropriate corpus for the domain. Our approach is applicable to narrow domains that may not have the huge training corpora typically used with word embedding approaches. In essence, by incorporating an external source of linguistic information, WordNet, and expanding the training corpus, we are getting more out of our training corpus.

For the future work, we plan to use our expanded corpus to train and evaluate the state of the art word embedding algorithms, such as BERT (*Devlin et al., 2019*), GPT-2 (*Radford et al., 2019*), CTRL (*Keskar et al., 2019*), and GPT-3 (*Brown et al., 2020*). Furthermore, we plan to use our collected ground truth datasets to evaluate the recent work done by Huang and his team that is called ClinicalBERT (*Huang, Altosaar & Ranganath, 2020*). Also, for the future work, we plan to do further improvements to the GloVeSyno, GloVeHypo, GloVeHyper algorithms and test them using the UMLS semantics to explore more laymen terms relationships. In our experiments, we implemented our algorithms on only unigram seed terms. We plan to explore applying these algorithms to different word grams of different lengths. Moreover, in our work, we used the *MedHelp.org* corpus to find new laymen term. Even though this corpus was rich with laymen information, our plan is to use larger healthcare dataset and might apply to multilanguage datasets to find laymen terms in different languages. In addition, we are currently exploring an iterative feedback approach to expand the corpus with words found by GloVe itself rather than those in an external linguistic resource. We are also working on our other project that tackle the problem of adding these laymen terms that laypeople use but not covered in the laymen vocabularies.

### Funding
The authors received no funding for this work.

### Competing Interests
Susan Gauch is an Academic Editor for PeerJ.

### Author Contributions
- Mohammed Ibrahim conceived and designed the experiments, performed the experiments, analyzed the data, performed the computation work, prepared figures and/or tables, authored or reviewed drafts of the paper, and approved the final draft.
- Susan Gauch conceived and designed the experiments, performed the experiments, analyzed the data, performed the computation work, prepared figures and/or tables, authored or reviewed drafts of the paper, and approved the final draft.
- Omar Salman analyzed the data, prepared figures and/or tables, authored or reviewed drafts of the paper, and approved the final draft.
- Mohammed Alqahtani analyzed the data, prepared figures and/or tables, authored or reviewed drafts of the paper, and approved the final draft.

### Data Availability
The data and code are available at the Text Analysis Research Lab: http://text.csce.uark.edu/chv.html.

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
