# Peer review of "An automated method to enrich consumer health vocabularies using GloVe word embeddings and an auxiliary lexical resource"

_PeerJ Computer Science, doi:10.7717/peerj-cs.668_

## Round 0.1 · original submission · Minor Revisions

Authors should revise paper and provide point-to-point responses according to the comments raised by the reviewers.

Reviewer 1 ·

Basic reporting

See general comments

Experimental design

See general comments

Validity of the findings

See general comments

Additional comments

The topic of consumer health vocabulary enrichment is interesting. The followings are issues needed to be addressed.

 the significance of the work should be highlighted at the beginning.
 justification about the use of similarity metric.
 comparisons of the similarities and differences in different algorithms used should be provided, and also link this with the results.
 What are the research and practice implications? This should be included in the discussion section.
 Most references are out-of-date, that is, published five years ago. More recent works in this field (that is, published in recent five years, particularly 2020-2021) should be added. Also, some references have incomplete compilation, e.g., missing volume/issue/page numbering.
 double-check both definition and usage of acronyms: every acronym should be defined only once (at the first occurrence) and always used afterwards (except for the abstract). There are mistakes in this issue. For example, NLP.
 the manuscript presents some bad English constructions and grammar mistakes: a professional language editing and careful proofread is strongly needed to sufficiently improve the paper's presentation quality.

Reviewer 2 ·

Basic reporting

The article describes an automatic method to enrich consumer health vocabularies such as OAC CHV and MedLine Plus, through GloVe word embeddings and wordNet. It is an extension of previous work from [1], including additional datasets for evaluation, but it should be detailed.
The article uses an unambiguous text, but there are a few typos:
• Line 95: vice versa, These
• Line 122: refers instead of refer.
• Some double spaces between lines 322 and 335.
• Use of the CHV acronym on the abstract but not on the text.
The article structure is also great, but some of the pictures seem not to be very relevant (3a and 3b)

[1] Ibrahim, M., Gauch, S., Salman, O., & Alqahatani, M. (2020). Enriching consumer health vocabulary using enhanced GloVe word embedding. arXiv preprint arXiv:2004.00150.

Experimental design

The article perfectly fits the scope of the journal. It contains a very complete state of the art with a very rigorous and technical investigation following relevant technologies.
The article is an extension of previous work including additional datasets for evaluation, but it must be detailed to clarify the new points.

Validity of the findings

No comment.

Additional comments

The article presents an automated method for enriching consumer health vocabularies through GloVe and wordNet word embeddings. It is a great investigation perfectly documented in this article, as it is shown on the quality of the state of the art and the good results obtained. I should only note a few small comments for publication, as well as suggest the possibility of using the UMLS semantic types to improve the semantics of the extensions.

---

## Round 0.2 · accepted · Accept

Congratulations on the acceptance of your manuscript. Well done!